# A Serosurvey of Japanese Encephalitis Virus in Monkeys and Humans Living in Proximity in Thailand

**DOI:** 10.3390/v15051125

**Published:** 2023-05-09

**Authors:** Divya Lakhotia, Yin May Tun, Nanthanida Mongkol, Oranit Likhit, Sarocha Suthisawat, Suthee Mangmee, Daraka Tongthainan, Wirasak Fungfuang, Phitsanu Tulayakul, Kobporn Boonnak

**Affiliations:** 1Department of Immunology, Faculty of Medicine Siriraj Hospital, Mahidol University, Bangkok 10700, Thailand; 2Division of Microbiology and Parasitology, Faculty of Medicine, Siam University, Bangkok 10160, Thailand; 3Faculty of Veterinary Medicine, Rajamangala University of Technology Tawan-ok, Chonburi 20110, Thailand; 4Department of Zoology, Faculty of Science, Kasetsart University, Bangkok 10900, Thailand; 5Department of Veterinary Public Health, Faculty of Veterinary Medicine, Kasetsart University, Bangkok 10900, Thailand

**Keywords:** Japanese encephalitis virus, seroprevalence, Plaque Reduction Neutralization Test, *Flavivirus*

## Abstract

Japanese encephalitis virus (JEV) is a member of the Flaviviridae family and one of Asia’s most common causes of encephalitis. JEV is a zoonotic virus that is transmitted to humans through the bite of infected mosquitoes of the *Culex* species. While humans are dead-end hosts for the virus, domestic animals such as pigs and birds are amplification hosts. Although JEV naturally infected monkeys have been reported in Asia, the role of non-human primates (NHPs) in the JEV transmission cycle has not been intensively investigated. In this study, we demonstrated neutralizing antibodies against JEV in NHPs (*Macaca fascicularis*) and humans living in proximity in two provinces located in western and eastern Thailand by using Plaque Reduction Neutralization Test (PRNT). We found a 14.7% and 5.6% seropositive rate in monkeys and 43.7% and 45.2% seropositive rate in humans living in west and east Thailand, respectively. This study observed a higher seropositivity rate in the older age group in humans. The presence of JEV neutralizing antibodies in NHPs that live in proximity to humans shows the occurrence of natural JEV infection, suggesting the endemic transmission of this virus in NHPs. According to the One Health concept, regular serological studies should be conducted especially at the animal–human interface.

## 1. Introduction

Japanese encephalitis virus (JEV) is a member of the *Flavivirus* genus in the *Flaviviridae* family, which also consists of dengue and Zika viruses [1]. JEV is a mosquito-borne zoonotic virus that is transmitted through the bite of vector *Culex* mosquitos, mainly *Culex tritaeniorhynchus*. It was first discovered and described in 1871 in Japan but the virus was isolated later in 1934 [2]. JEV is endemic to Southeast Asia and the Western Pacific region and it is estimated that more than 3 billion people are at risk of disease. A systematic review conducted by the WHO in 2011 states that approximately 68,000 cases occur annually in the endemic region [3]. Recent studies evaluating JEV disease burden estimated around 57,000 cases and 21,000 deaths in 2019 alone [4].

JEV is endemic to Thailand and has been the cause of several encephalitis cases; approximately 2500 cases were reported annually during the 1970s and 1980s [5]. In the northern part of the country, JE resulted in significant childhood morbidity, causing up to 2400 cases and 400 deaths annually [6]. After the introduction of the national vaccination program for JEV in 1990, there was a significant reduction in cases [5]. Since 2016, the three-dosed mouse-brain-derived JEV vaccine was replaced by two doses of live attenuated JEV vaccine, administered at 1 years followed by 2.5 years of age [7,8]. The latest report by the Ministry of Public Health (MOPH), Thailand, states that, between the years 2009 to 2020, the annual JEV-infected cases ranged from 8 to 61 cases nationally [9]. However, the lack of specific serological tests makes distinguishing between JEV infection and infection by another flavivirus in an endemic area difficult. Other important viruses in the Flaviviridae family that are endemic to Thailand include dengue serotype 1–4 and Zika virus, thus resulting in an underestimate of the actual number of cases in the country [1].

Domestic animals such as pigs and water birds are studied extensively for their role in JEV transmission. They can act as important reservoirs for the virus because infection leads to a high viremia load in these animals, thus allowing more uninfected mosquitos to be exposed to the virus and continuing the transmission cycle. Humans are dead-end hosts for JEV because the viral load upon infection is not high enough to cause transmission to other mosquitos [10]. In contrast to domestic animals, the role of wild vertebrates such as monkeys in the JEV transmission cycle has not been intensively studied. There have been a few studies reporting the seroprevalence of JEV in long-tailed macaques (*Macaca fascicularis*) with varying results [11,12,13]. The presence of antibodies in these long-tailed macaques indicates natural infection. However, the role of NHPs in JEV transmission must be carefully characterized. It is still unclear whether the JEV viremia in NHPs is sufficient to be infectious to mosquito vectors. Therefore, like humans, NHPs may act as a dead-end host for JEV. The role of NHPs has also been studied in the spread of other viruses, such as Zika and chikungunya virus. Arboreal mosquitos participate in the sylvatic cycle, which sometimes leads to spillover events, encroaching on the urban cycle and affecting human populations [14,15,16]. However, such studies on the spread of JEV are limited and a study of JEV seropositivity in both wild monkeys and humans living in proximity has not been investigated. Because JEV is one of the leading causes of encephalitis in Asia [3], understanding the role of wild animals in proximity to humans is crucial to controlling disease transfer. In this study, we conduct a serosurvey of JEV antibodies in long-tailed macaques (*Macaca fascicularis*) and humans living in proximity in two provinces of Thailand. According to the One Health concept [17], this kind of surveillance is vital, especially in JEV endemic areas.

## 2. Materials and Methods

### 2.1. Ethical Approval

The use of human samples in this study was approved by the ethics committee of the Faculty of Tropical Medicine, Mahidol University, Thailand, under approval number FTM-ECF-020-04.

The use of monkey samples was approved by the Institutional Animal Care and Use Committee of Mahidol University, Thailand, and Kasetsart University under approval number AUKU60-VET-049 and FTM-ACUC003/2019 E.

### 2.2. Monkey Blood Samples

In this study, 210 blood samples were collected from long-tailed macaques (*Macaca fascicularis*) by the Department of National Park, Wildlife and Plant Conservation, Thailand, and Kasetsart University. One hundred and two monkey blood samples were collected from Prachuap Khiri Khan (study site 1) in July 2017 and 108 monkey blood samples were collected from Chonburi (study site 2) in August 2018 (Table 1). There is no report of JEV serology assessment in humans during this time frame. However, according to the ministry of public health Thailand, there were 14 and 18 JEV-confirmed cases in 2017 and 2018, respectively [9]. The monkeys were captured as previously described [18]. Gender and anthropological measurements were taken (weight, arm length, tail length, and body length). Dental casts and dental photographs were also taken. Monkeys were bled from the inguinal vein while sedated. All blood samples were centrifuged at 1500 RPM for 10 min to collect blood sera, which were stored at −20 °C for further experiments.

### 2.3. Human Blood Samples

We recruited and enrolled Thai adults at Prachuap Khiri Khan (study site 1) in October 2019 and Chonburi (study site 2) in January 2020. Eligible participants were healthy adults between the ages of 18 and 55 years old who lived or worked within 5 km of the monkey’s habitat. All participants provided written informed consent before enrolment. All participants agreed to fill out questionnaires pertaining to lifestyle choices relevant to the study, such as vicinity to urban or rural areas and occupation. Vaccination history was assessed but only 5 participants had documented history of receiving JEV vaccination. Blood samples were centrifuged at 1500 RPM for 10 min to collect serum samples, which were kept at −20 °C for further experiments.

### 2.4. Virus Propagation

JEV strain SA-14-14-2 was kindly provided by Dr. Thaneeya Duangchinda (Division of Dengue Hemorrhagic Fever Research, Faculty of Medicine Siriraj Hospital, Mahidol University). The virus was further propagated in Vero cells, kindly provided by Dr. Stephen Whitehead (NIAID, NIH, USA). Briefly, the virus stock was diluted to 1:10 and added to a T75 flask containing 70–80% confluent Vero cells at 37 °C with 5% CO_2_ for 60 min, followed by adding 15 mL of growth medium. The infected cells were then incubated at 37 °C with 5% CO_2_ for 6 days. After cell cytopathic effect (CPE) was observed in the monolayer, JEV-containing supernatant was collected, aliquoted, and stored at −80 °C for further experiments.

### 2.5. Virus Titration

Plaque titration was performed in a 24-well plate in duplicate wells to determine the virus concentration in terms of plaque-forming unit per mL (PFU/mL). Briefly, virus stock was 10-fold serial diluted with Opti-MEM (Invitrogen, New York, NY, USA). Cell culture medium was removed from 70–80% confluent monolayer of Vero cells on 24-well plates, and 105 µL of the diluted virus was added to each well. The cell monolayers were incubated at 37 °C with 5% CO_2_ for 60 min, then overlaid with 0.5% methylcellulose in MEM supplemented with 2% Fetal Bovine Serum (FBS) (Gibco, Waltham, MA, USA). The infected cell monolayers were further incubated at 37 °C with 5% CO_2_ for 7 days. Methylcellulose was removed from infected cell monolayers and the plaques were visualized by staining with 0.2% crystal violet (Sigma-Aldrich, St. Louis, MO, USA). Virus concentration was calculated based on dilution factor and volume of diluted virus per well.

### 2.6. Plaque Reduction Neutralization Test (PRNT)

Plaque Reduction Neutralization Test (PRNT) was performed as previously described [19]. Briefly, serum samples were heat inactivated at 56 °C for 30 min, and serial 4-fold dilution beginning at 1:5 was performed with Opti-MEM supplemented with heat-inactivated FBS. JEV diluted to a final concentration of 1000 PFU/mL was added to equal volumes of diluted serum, mixed well and incubated at 37 °C with 5% CO_2_ for 30 min. Cell culture medium was removed from 70–80% confluent Vero cell monolayers on 24-well plates, and 105 µL of the virus–serum mixture was transferred onto duplicate wells of cell monolayers. Cell monolayers were incubated for 60 min at 37 °C with 5% CO_2_ and overlaid with 0.5% methylcellulose in MEM supplemented with 2% FBS. Cell monolayers were further incubated at 37 °C with 5% CO_2_ for 7 days. The plaques were visualized by staining with 0.2% crystal violet (Sigma-Aldrich). Using a curve-fitting method, the PRNT titer was calculated based on a 50% reduction in plaque counts (PRNT_50_) [19]. PRNT_50_ = 5 was the lower limit of antibody detection and a titer level of PRNT_50_ ≥ 10 was considered to be positive for the presence of JEV-neutralizing antibody, as described in prior studies [20].

### 2.7. Statistical Analysis

All statistical analysis was performed using IBM SPSS Version 26.0 and GraphPad PRISM Version 9.3.1. Descriptive statistics were used to create the characteristics of the study population. Mann–Whitney and one-way ANOVA tests were performed to compare the geometric mean titer (GMT) of PRNT_50_ amongst different study sites in monkey and human samples and age groups in human samples, respectively. Univariate analysis was performed to determine the factors associated with seropositivity in human samples, and the odds ratio was calculated. *p*-values < 0.05 were considered statistically significant for all statistical tests. The map of Thailand indicating study sites was created using the QGIS software Version 3.26.

## 3. Results

### 3.1. Characteristics of the Study Population

One hundred and two monkey serum samples were collected from study site 1 (Figure 1). Among these samples, 1 out of 102 monkey serum samples was collected from a juvenile monkey (Table 1). For study site 2, 20 out of 108 samples were collected from juvenile monkeys. Most of the samples collected from study site 1 were from male monkeys; 86 out of 102 (84.3%) and all 108 samples out of 108 (100%) collected from study site 2 were from male monkeys. Study site 1 is the forest adjacent to the tourist attraction area with a mountain along the beach, whereas study site 2 is a peri-urban area where the monkeys live near the port.

The characteristics of the human samples are summarized in Table 2. For study site 1, 14 (11.1%), 39 (31.0%), 37 (29.4%) and 36 (28.6%) were from the <24, 24–35, 36–45 and older than 45 age group, respectively. As for study site 2, 12 (10.4%), 37 (32.2%), 21 (18.3%) and 45 (39.1%) were from the <24, 24–35, 36–45 and older than 45 age group, respectively. The majority of participants were female in both study sites; 85 out of 126 (67.5%) in site 1 and 80 out of 115 (69.6%) in site 2.

### 3.2. Seropositivity of JEV Antibodies in Monkeys

Amongst the 102 monkey samples collected from study site 1, 15 (14.7%) were seropositive for JEV antibodies (Appendix A). Ten out of fifteen of the positive samples had a PRNT_50_ titer > 20 with a maximum antibody titer of 310.7. Out of 108 monkey samples in study site 2, 6 (5.6%) were seropositive for JEV antibodies. Four out of six of the positive samples had a PRNT_50_ titer > 20 with a maximum antibody titer of 109.7. The GMT of JEV antibodies in monkey samples was 7.45 (95% CI: 6.10–9.09) in study site 1 and 5.58 (95% CI: 5.08–6.12) in study site 2 with a significant difference (Figure 2). There was no significant association between gender and seropositivity in study site 1 or age group and seropositivity in study site 2.

### 3.3. Seropositivity of JEV Antibodies in Humans

The result of seropositivity by PRNT_50_ in humans is shown in Appendix A. Study site 1 contained 55 out of 126 (43.7%) seropositive samples, while study site 2 contained 52 out of 115 (45.2%) seropositive samples. In site 1, 39 out of 55 positive samples had a PRNT_50_ titer > 20 with a maximum antibody titer of 548.3. In site 2, 35 out of 52 positive samples had a PRNT_50_ titer > 20 with a maximum antibody titer of 680.5. The GMT of JEV antibodies in human samples was 12.68 (95% CI: 10.31–15.60) in site 1 participants and 14.10 (95% CI: 11.04–18.01) in site 2 participants with no significant difference (Figure 3).

### 3.4. Factors Associated with JEV Seropositivity in Humans

Univariable analysis of factors that may be associated with JEV seropositivity in humans was conducted as shown in Table 3. Age group was the only factor significantly associated with JEV seropositivity; older age group was positively associated with JEV seropositivity. All other factors, including gender, house environment, and occupation, were not significantly associated with JEV seropositivity.

The association of age and JEV seropositivity was further studied by examining the seropositive proportion and GMT of JEV antibodies of different age groups. The seropositive proportion was 7 out of 26 (26.9%), 32 out of 76 (42.1%), 22 out of 58 (37.9%) and 46 out of 81 (56.8%) for the age groups <24, 24–35, 36–45 and older than 45, respectively (Figure 4). The GMT in human samples was 10.23 (95% CI: 6.61–15.82), 12.62 (95% CI: 9.54–16.69), 11.70 (95% CI: 8.49–16.12) and 16.81 (95% CI: 12.55–22.51) for the age groups <24, 24–35, 36–45 and older than 45, respectively. When GMT results were statistically analyzed, no significant difference was observed among different age groups.

## 4. Discussion

The role of animals such as pigs and birds in JEV transmission is already well established. However, our understanding of this zoonotic virus is still incomplete. Since the presence of JEV antibodies in NHPs is clearly reported in previous studies [11,12,13], we investigate seropositivity against JEV in monkeys at an animal–human interface in Thailand. We chose two provinces close to the capital Bangkok with a high density of monkeys living in proximity to human populations for this study. Study site 1, Prachuap Khiri Khan, is a popular tourist attraction spot, whereas study site 2, Chon Buri, is an important industrial area. The presence of JEV antibodies in monkeys suggests ongoing transmission of JEV in NHPs in both study sites. In agreement with JEV seroprevalence in monkeys (*Macaca fascicularis*) in Malaysia [13] and the study in captive monkeys (*Macaca nemestrina*) in Northern Thailand [21], the seropositivity rate observed from monkeys living in study site 1 was 14.3%. However, studies conducted on monkeys (*Macaca fascicularis*) in Indonesia and the Philippines had a higher seropositivity rate of 41.3% and 35.2%, respectively [11,12]. Even captive monkeys (*Macaca fuscata*) in Japan showed a higher seropositivity rate of 44% [22]. On the other hand, seropositivity rate in monkeys living in study site 2 is significantly lower than that observed in study site 1. Our results indicate that monkeys in study site 1, the Prachuap Khiri Khan province, have a greater history of JEV infection than those in the Chon Buri province (study site 2). The low seropositivity amongst monkeys in the Chon Buri province could be explained by their location. Monkeys in this region are usually found around the Laem Chabang port, where our samples were collected. This is an industrial area and is not as crowded when compared to Prachuap Khiri Khan. The difference in environment may result in a lower vector density or different vector species in Laem Chabang.

The presence of antibodies against JEV in monkey samples confirms the presence of natural infection in monkey populations in this region. The sylvatic cycle between monkeys and mosquitos has been observed in other flaviviruses, such as dengue and yellow fever [16,23]. Therefore, according to the One Health Concept, constant and vigilant studies need to be conducted to identify the role of monkeys in JEV transmission. This is especially important in areas where humans live in tandem with wildlife.

For JEV serosurvey in the human population living in proximity to NHP habitats, there was no significant difference between the two study sites, indicating that both have a similar seropositivity rate, and less than half the population has neutralizing antibodies against JEV. Our result is similar to that observed in the people of Chiang Mai, a JEV endemic area in the north of Thailand [7]. When compared to other countries in the JEV endemic area, the seropositivity rate in our study is much lower than in Korea. Korea has one of the highest rates of seroprevalence out of all the JEV endemic countries, between 97.8 and 98.3% for all age groups [24,25]. This is attributable to two specific reasons: high incidence of JE cases with a large magnitude of epidemics and a mandatory vaccination program in children [26]. The low seropositive rate in our study could be due to the lack of natural infection, low vaccine coverage, and ineffectiveness of the vaccines in Thailand [27].

When we examined the factors associated with seropositivity in humans, we found age to be significantly associated. Unlike the observation in Chiang Mai, where the highest seropositivity rate was observed in the adolescent group (10–20 years) [7], we found that JEV seropositivity in our participants correlated with increasing age group. The result in Chiang Mai is comparable to a nationwide study conducted in Taiwan, which had a seroprevalence rate of 74%, 63%, 55%, 54%, 68% and 86% in the age group 16–21, 22–26, 27–32, 33–39, 40–49 and 50–90, respectively [28]. The result discrepancy between our study and the one in Taiwan may have ensued from a difference in the study population, living style and coverage of vaccination.

A possible explanation for our results is that probability of JEV exposure increases with age [29] and, since natural infection is more likely to produce a higher antibody titer with a longer-lived immune response than vaccination, those older than 45 have the highest rate of seropositivity out of all the age groups [30].

This study has some strengths, especially the use of PRNT_50_ for the detection of neutralizing antibodies in human and monkey samples. PRNT_50_ is considered to be the gold standard for differential serodiagnosis of flaviviruses [31]. This is important because other diagnostic tests, such as ELISA, often result in cross-reactivity between flavivirus [32].

This study also has some limitations. We were able to procure only one juvenile monkey sample from study site 1, while site 2 consisted of male monkey samples only. This meant that we were not able to conclusively study the association of age group and gender in seropositivity in monkeys. We were also unable to obtain accurate JEV vaccination history of the human participants and the PRNT_50_ assay could not discriminate the antibody responses from vaccination or natural infection. Thus, we could not account for that in the seropositivity of the samples. Although PRNT_50_ is the gold standard for determining JEV-neutralizing antibodies, the serological cross-reactivity among flaviviruses, especially DENV, remains a concern. It is vital to determine the seropositivity rate of arboviruses such as DENV and ZIKV in NHPs living in proximity to human populations. The current transmission ecology of these viruses in Southeast Asia still needs to be characterized. There is a gap in our understanding of the fundamental biology of these arboviruses in Southeast Asia. Additionally, monkey and human samples were not collected in the same year, which could affect the relationship we are trying to assess in our study. Lastly, the age range of our study participants was quite narrow and we were not able to study seropositivity in children and the elderly.

In conclusion, this study shows the presence of JEV-neutralizing antibodies in monkeys from two central provinces in Thailand where they live in proximity to humans. The presence of JEV-neutralizing antibodies in NHPs mainly informed that the NHPs could be one of the alternative hosts. Still, the definite role of NHPs in JEV transmission cycles warrants further intensive studies. For instance, virus isolation and RNA detection from monkey samples, followed by sequencing, should be performed to check for genetic markers relevant to virus fitness. Additionally, the prevalence of *Culex* mosquitos in these study sites should be investigated, including feeding patterns and infectious doses relevant to NHPs.

According to the One Health concept, routine serological surveys are crucial for managing infectious diseases. Serologic surveillance in pigs has been utilized to anticipate JEV outbreaks in people. However, several studies, including our study, have suggested that other species, such as NHPs, should also act as sentinel animals. As such, routine serological surveys in NHPs should take place, especially in areas where monkeys live proximal to the human population, such as Southeast Asia [33]. Additionally, prevalence of *Culex* mosquitos in these study sites should be investigated, including feeding patterns and infectious dose relevant to NHPs. Lastly, when the factors associated with seropositivity in humans were assessed, we found older age to be significantly associated with increased seropositivity. Hence, vaccination in this age group might also need to be considered, not just in children.

## Figures and Tables

**Figure 1 viruses-15-01125-f001:**
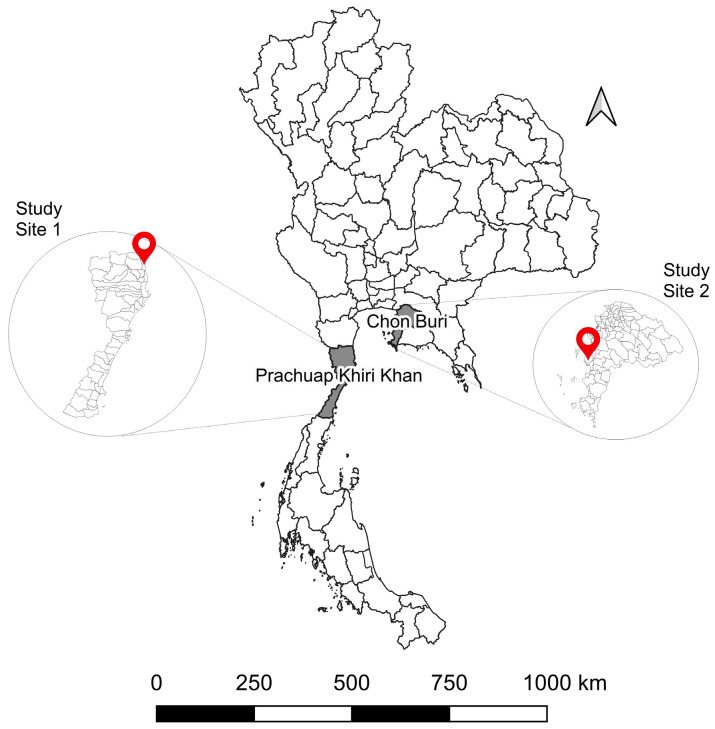
Geographical location of sample collection sites in Thailand. (Study site 1) Kao Takiab, Prachuap Khiri Khan (GPS: 12.515337, 99.982201). (Study site 2) Laem Chabang, Chon Buri (GPS: 13.081948, 100.881946).

**Figure 2 viruses-15-01125-f002:**
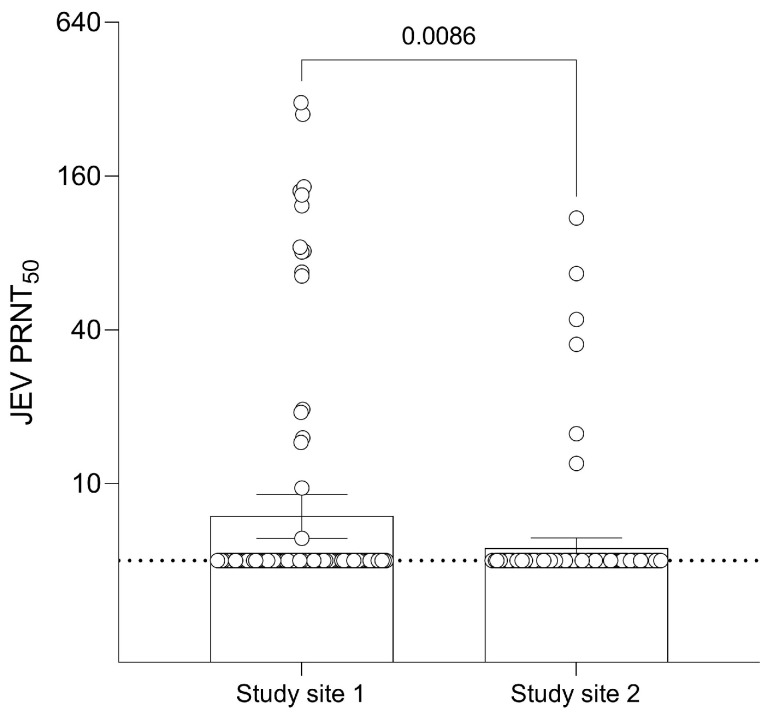
Geometric Mean Titer (GMT) of 50% Plaque Reduction Neutralization Test (PRNT_50_) levels for JEV antibodies of different sites in monkeys. Dashed line at JEV PRNT_50_ = 5 represents the lower limit of antibody detection. Error bars represent 95% CI for the GMT.

**Figure 3 viruses-15-01125-f003:**
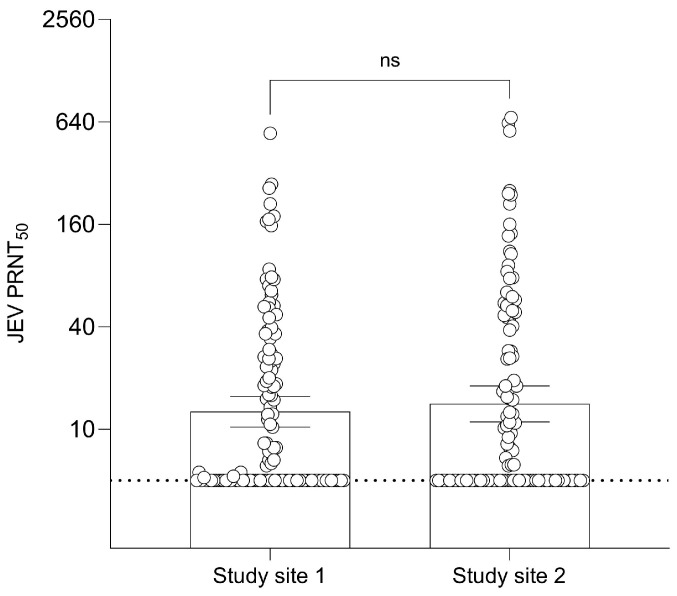
Geometric Mean Titer (GMT) of 50% Plaque Reduction Neutralization Test (PRNT_50_) levels for JEV antibodies of the two different sites in humans. Dashed line at JEV PRNT_50_ = 5 represents the lower limit of antibody detection. Error bars represent 95% CI for the GMT.

**Figure 4 viruses-15-01125-f004:**
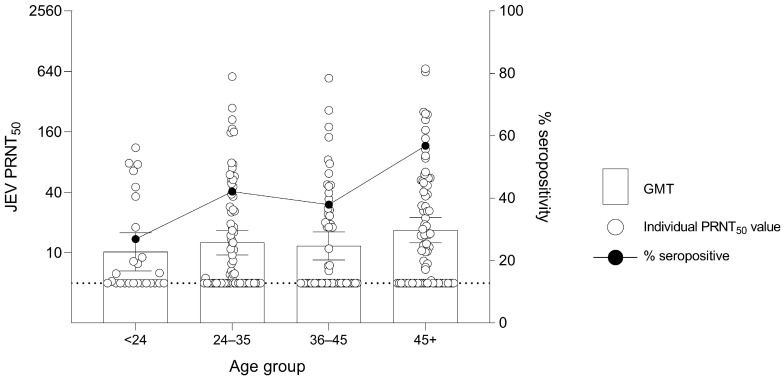
Geometric Mean Titer (GMT) of 50% Plaque Reduction Neutralization Test (PRNT_50_) levels for JEV antibodies and percentage seropositivity of different age groups in humans. Dashed line at JEV PRNT_50_ = 5 represents the lower limit of antibody detection. Error bars represent 95% CI for the GMT.

**Table 1 viruses-15-01125-t001:** Characteristics of the monkey study population.

Characteristics ^1^	Study Site 1 (*n* = 102)	Study Site 2 (*n* = 108)
Age group		
Juvenile	1 (1.0%)	20 (18.5%)
Adult	101 (99.0%)	88 (81.5%)
Gender		
Male	86 (84.3%)	108 (100%)
Female	16 (15.7%)	0 (0%)

^1^ Data are presented as *n* (% of total) for all the variables.

**Table 2 viruses-15-01125-t002:** Characteristics of the human study population.

Characteristics ^1^	Study Site 1 (*n* = 126)	Study Site 2 (*n* = 115)
Age group, in years		
<24	14 (11.1%)	12 (10.4%)
24–35	39 (31.0%)	37 (32.2%)
36–45	37 (29.4%)	21 (18.3%)
45+	36 (28.6%)	45 (39.1%)
Gender		
Male	41 (32.5%)	35 (30.4%)
Female	85 (67.5%)	80 (69.6%)
House environment		
Urban	102 (81%)	90 (78.3%)
Near a garden	10 (7.9%)	6 (5.2%)
Other	14 (11.1%)	19 (16.5%)
Occupation		
Unemployed	10 (7.9%)	16 (13.9%)
Government/Private	29 (23.0%)	23 (20.0%)
Business	27 (21.4%)	25 (21.7%)
Contractor	46 (36.5%)	42 (36.5%)
Others	14 (11.1%)	9 (7.8%)

^1^ Data are presented as *n* (% of total) for all the variables.

**Table 3 viruses-15-01125-t003:** Univariable analysis of factors associated with JEV seropositivity amongst human samples.

Characteristic	OR (95% CI)	*p*-Value
Age group, in years		
<24 ^1^	1	
24–35	1.974 (0.742–5.254)	0.173
36–45	1.659 (0.601–4.582)	0.329
>45	3.567 (1.350–9.427))	0.010 ^2^
Gender		
Female (vs. Male)	1.728 (0.929–3.216)	0.084
House environment		
Urban ^1^	1	
Near a garden	1.894 (0.602–5.952)	0.275
Others	0.817 (0.354–1.889)	0.637
Occupation		
Contractor ^1^	1	
Unemployed	0.759 (0.292–1.970)	0.571
Government/Private	1.046 (0.485–2.253)	0.909
Business	1.260 (0.596–2.663)	0.546
Others	0.878 (0.242–3.185)	0.843

Abbreviations: OR, odds ratio; CI, confidence interval. ^1^ Reference group; ^2^ statistically significant association (*p* < 0.05).

## Data Availability

Data is contained within the article and supplement material.

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
