# Peer review of "A Serosurvey of Japanese Encephalitis Virus in Monkeys and Humans Living in Proximity in Thailand"

_viruses, 2023, doi:10.3390/v15051125_

Round 1

Reviewer 1 Report

The authors report seroprevalence data that long-tailed macaques from two particular sites in Thailand are exposed to JEV and develop detectable neutralizing antibody titers. However, they make a rather big jump in conclusion and suggest that NHPs may have a potential role on the sylvatic cycle of JEV as "intermediate" or "secondary or amplification hosts." From what is reported in literature, NHPs do not produce high enough viremia to infect feeding mosquitoes and, thus, are dead-end hosts that are not very significant players in terms of maintaining the transmission cycle of JEV. The authors did not find any evidence in terms of detecting high viral RNA or infectious titer in the monkey blood samples to support this claim - only serology data. Additionally, they also report some serology and population data on some humans that were sampled nearby those two particular sites, but at different years. The data produced in this study can be helpful for JEV research, but the dialogue/presentation of data/conclusions must be modified appropriately.

Below are more recommendations to improve the manuscript:

1) Line 45: What are the other flaviviruses endemic to Thailand? Please include to provide more information to the readers.

2) Line 56: To determine the possibility of these monkeys to act as amplification hosts, their ability to develop high enough viremia AND that competent mosquitoes (in significant numbers/preference, etc) can become infected upon feeding on the viremic monkeys must be demonstrated. 

3) Section 2.2 Monkey blood samples: (a) Are there any accompanying seroprevalence data or reported cases of infected/exposed humans in 2017 and 2018? Whether or not the information is available, please include it in this section. (b) Please also provide details of the season/months or period of time in which the monkey and human samples were collected. (c) Are there any mosquito surveillance data you can pair it up with these timepoints? It would buff up the information. 

4) Section 2.3 Human blood samples: It is stated in the introduction that there is a national vaccination program for JEV in Thailand. Is this a mandatory program? What types of vaccines are offered? Please include in this section whether or not vaccination status/history/type of vaccine was part of the questionnaire. 

5) Figure 1: Please add the labels "Site 1" and "Site 2" to the figure.

6) Line 281: Please provide more explanations to why mostly adult and male monkeys were captured. Is it the capturing technique? Does the technique unintentionally target that type of population of monkeys? Season variation? When were the monkey samples collected?

7) Line 289: The role of monkeys can be further investigated through viral RNA detection and quantification and the inclusion of the mosquito vectors (feeding pattern/preferences, susceptibility? infectious dose? etc.) 

Author Response

Response to Reviewer 1

Reviewer 2 Report

This paper investigates the prevalence of JEV in NHPs (Macaca fascicularis) and nearby residents in two provinces of western and eastern Thailand. The presence of JEV neutralizing antibodies in NHPs living with humans was found, suggesting an important influence of NHPs in the transmission of JEV. This may be of interest to researchers in related fields, but the paper in its current state is diffuse and lacks focus or firm conclusions.

1. The abstract of Age was the factor significantly associated with increased seropositivity in humans univariable analysis is not supported by detailed data in the results section.

2Please make it clear for the correspondence of site 1/2 in text and site A/B in Figure 1 and tables.

3. Part of Characteristics of the study population should provide a brief description of the inhabitants of the survey area

4. It is suggested that Table 3 be presented separately for clarity as 3.2 and 3.3 are two parts.

5. The result of “age group was the only factor significantly associated with JEV seropositivity” .The sample number of people aged <24 is not enough to support this conclusion.

6.This article is a retrospective study, would it be possible to provide advice on the prevention of JEV from the One Health perspective at the end of the article?

Author Response

Response to Reviewer 2

Reviewer 3 Report

The manuscript addresses an important concern: seroprevalence of neutralizing antibodies against Japanese encephalitis virus in monkeys (Macaca fascicularis) and humans living in Thailand. In order to achieve the purposes of the study, the authors used the PRNT50 technique. The results are clearly presented but several concerns have to be addressed:

1. Abstract. Line 15: Please considerer writing “JEV” instead of “JE”.

2. Abstract. Lines 22-24: Regarding the experimental design of the study, I see that the data do not support this suggestion.

3. Introduction. Lines 35-36: Wouldn't there be more up-to-date data? Furthermore, I find it interesting to add the number of severe clinical cases or estimates of deaths from the disease.

4. Materials and Methods. The authors talk about seroprevalence, but was there a sample size calculation to infer that the data represent prevalence? If this has not been done, I see that the data are more related to JEV seropositivity.

5. Results. Figure 1: I suggest replacing A and B in the figure with 'study site 1' and 'study site 2', respectively.

6. Results. Table 3: There is no need to place this table, as the data were described in sections 3.2 and 3.3. 

7. Discussion. The discussion can be further enriched, as I see that other study addressing the seroprevalence of JEV in monkeys were not used, such as: doi, 10.1007/s10329-014-0421-7.

8. Discussion. I see that it could address more the diagnostic bias that may have occurred due to the loss of vaccination history in humans.

9. It would be interesting for the authors to describe future perspectives in exploring other endemic flaviviruses that may generate cross-reactivity of antibodies by serological tests. Again this is another factor that can cause diagnostic bias.

Author Response

Response to Reviewer 3

Round 2

Reviewer 1 Report

The authors have included the appropriate edits and provided good explanations to my comments. The manuscript reads more complete. Significance of the paper was also elevated by including the comments of One Health. There are just two minor edits:

1) Line 296: ZIKV is mentioned twice in the list.

2) Line 309: Italicize "Culex."